# Specific Cellular and Humoral Response after the Third Dose of Anti-SARS-CoV-2 RNA Vaccine in Patients with Immune-Mediated Rheumatic Diseases on Immunosuppressive Therapy

**DOI:** 10.3390/biomedicines11092418

**Published:** 2023-08-29

**Authors:** Kauzar Mohamed Mohamed, María Paula Álvarez-Hernández, Carlos Jiménez García, Kissy Guevara-Hoyer, Dalifer Freites, Cristina Martínez Prada, Inés Pérez-Sancristóbal, Benjamín Fernández Gutiérrez, Gloria Mato Chaín, Maria Rodero, Antonia Rodríguez de la Peña, Teresa Mulero, Cecilia Bravo, Esther Toledano, Esther Culebras López, Beatriz Mediero Valeros, Pedro Pérez Segura, Silvia Sánchez-Ramón, Gloria Candelas Rodríguez

**Affiliations:** 1Department of Immunology, IML and IdISSC, Hospital Clínico San Carlos, Calle Profesor Martín Lagos, S/N, 28040 Madrid, Spain; 2Rheumatology Department, Hospital Universitario Clínico San Carlos, Universidad Complutense de Madrid, 28040 Madrid, Spain; 3Unidad de Vacunación del Adulto, Servicio de Medicina Preventiva, Hospital Clínico San Carlos, 28040 Madrid, Spain; 4Department of Microbiology, IML and IdISSC, Hospital Clínico San Carlos, 28040 Madrid, Spain; 5Department of Medical Oncology, Hospital Clinico San Carlos, IdISSC, Calle Profesor Martín Lagos, 28040 Madrid, Spain

**Keywords:** SARS-CoV-2, T cell response, booster, IMRDs, humoral response

## Abstract

Objective: Data on cellular and humoral immunogenicity after the third dose of anti-SARS-CoV-2 vaccines in patients with immune-mediated rheumatic diseases (IMRDs) are scarce. Herein, we evaluated the adaptive immune response in IMRD patients treated with different immunosuppressive therapies (conventional synthetic disease-modifying antirheumatic drugs [csDMARDs], biological disease-modifying antirheumatic drugs [bDMARDs], and targeted synthetic disease-modifying antirheumatic drugs [tsDMARDs]) after the booster of the anti-SARS-CoV-2 vaccine to determine whether any drug reduced the vaccine’s response. Methods: A single-center prospective study was conducted, including patients presenting with IMRD and healthy controls (HC). Specific anti-SARS-CoV-2 interferon-gamma (IFN-γ) production was evaluated between 8–12 weeks after the third dose of the SARS-CoV-2 vaccine. In addition, anti-Spike IgG antibody titers were also measured. Results: Samples were obtained from 79 IMRD patients (51 women, 28 men; mean age 57 ± 11.3 years old): 43 rheumatoid arthritis, 10 psoriatic arthritis, 14 ankylosing spondylitis, 10 undifferentiated spondyloarthritis, and 2 inflammatory bowel disease-associated spondyloarthritis (IBD-SpA). In total, 31 HC (mean age 50.9 ± 13.1 years old, 67.7% women) were included in the study. Post-vaccine results displayed positive T-cell immune responses in 68 out of 79 (86.1%) IMRD patients (82.3% of those without prior COVID-19). All HC and IMRDs patients had an antibody response against the SARS-CoV-2 receptor-binding domain; however, the HC response was significantly higher (median of 18,048 AU/mL) than in IMRDs patients (median of 6590.3 AU/mL, *p* < 0.001). MTX and leflunomide were associated with lower titers of IgG and IFN-γ responses. Among bDMARDs, adalimumab, etanercept, and guselkumab are associated with reduced cellular responses. Conclusion: Our preliminary data show that the majority of our IMRD patients develop cellular and humoral responses after the SARS-CoV-2 booster vaccination, emphasizing the relevance of vaccination in this group. However, the magnitude of specific responses was dependent on the immunosuppressive therapy administered. Specific vaccination protocols and personalized decisions about boosters are essential for these patients.

## 1. Introduction

Faced with the current COVID-19 pandemic, an urgent need has arisen to understand the impact of immunosuppressive therapies on the efficacy of SARS-CoV-2 vaccines. Patients with immune-mediated rheumatic diseases (IMRDs) have been considered a high-risk group for severe disease, either because of the disease itself or the immunomodulatory therapies used. Studies conducted during the pandemic in this population suggest that these patients experience significantly higher rates of hospitalization, severe disease, morbidity, and mortality [1,2,3,4,5].

Clinical trials involved in the development of these vaccines were designed to measure the prevention of severe disease in the healthy population [6,7,8,9]. However, to date, these studies have not included patients receiving immunosuppressive therapy. The degree to which the immune response is altered may vary depending on the specific immunomodulatory regimen and vaccine used. Previous studies concerning other vaccines in patients with IMRDs, such as influenza or pneumococcus, have shown impaired humoral responses in these patients, especially in those who received rituximab treatment [10,11,12,13]. However, existing data derived from experience with other types of vaccines may not translate to the new vaccines implemented for COVID-19.

Initially, studies were conducted based on the humoral response in this group of patients, concluding that, in general, patients generate specific antibody responses depending on the therapeutic target of the immunosuppressive treatment and their intrinsic characteristics [14,15,16,17,18,19,20].

Although current vaccine efforts have focused on the induction of neutralizing antibodies against SARS-CoV-2, which wanes over time, there is a large body of evidence that the SARS-CoV-2-specific T cell response is essential for viral clearance, disease outcome, and long-term memory [21,22,23]. Experimental data suggest that CD8^+^ T-cell responses may play a protective role in the presence of decreasing or subprotective antibody titers [22,24]. Recently, new research studies have been conducted to evaluate the cellular response generated, showing that these vaccines are immunogenic, although the response is in some cases significantly lower compared to healthy controls [25].

Various hypotheses have been propagated as to what factors influence vaccine immunogenicity, such as the type of pharmacological immunosuppression or patient intrinsic factors [26,27,28]. Published data regarding immune responses identify drugs targeting B cells as risk factors for low immunogenicity [15,17,29,30]. However, there are variable results in studies of patients treated with these drugs, with some confirming that both humoral and cellular responses are diminished and others showing a cellular response that could confer immunogenicity in this subgroup of patients [31,32].

There are several types of drugs available for treating IMRDs, which fall under the category of disease-modifying antirheumatic drugs (DMARDs). The three sub-categories of DMARDs are conventional synthetic DMARDs (csDMARDs), which were the first drugs to slow the progression of the disease and induce remission; biologic DMARDs (bDMARDs), which consist of antibodies that target crucial inflammatory or immune pathways; and targeted synthetic DMARDs (tsDMARDs), which are inhibitors of Janus kinase (JAK). These molecules play a role in the signal transmission of inflammatory pathways that can help regulate immune cells and inflammation [33,34].

Given the need to implement vaccine strategies in these patients, our objective was to describe the serological and T-cell responses after the third dose of the vaccine in a cohort of patients with IMRDs (rheumatoid arthritis and spondyloarthropathies) treated with immunosuppressive therapy (csDMARDs, bDMARDs, and tsDMARDs). We aimed to identify the impact of these treatments on vaccine response and determine which patients would benefit from various vaccination strategies.

## 2. Materials and Methods

### 2.1. Study Design

This was a single-center prospective observational study. Seventy-nine adult IMRDs treated with different immunosuppressive therapies (csDMARDs, bDMARDs, and tsDMARDs) who received the third dose of the SARS-CoV-2 vaccine were studied. We measured the adaptive immune response two weeks after stopping treatment to reduce the influence of anti-inflammatory drugs in the assays. During the study, samples were collected according to the visits scheduled in the care of each patient to avoid extra visits to the hospital due to the pandemic situation. A group of sex- and age-matched healthy controls (HC), healthcare workers (HCWs), and nonclinical personnel with no known rheumatic diseases and receiving no immunosuppressive medications were used as a comparator group for the study (n = 31). Thus, blood samples were taken between 8–12 weeks after the third dose of the SARS-CoV-2 vaccination (BNT162b2 [Pfizer/BioNTech, New York, NY, USA] or mRNA-1273 [Moderna, MA, USA]). Between November 2021 and February 2022, we conducted a search in the electronic clinical database to collect information on age, gender, type of rheumatic diagnosis, pharmacological history, and COVID-19 vaccination. We also examined the patients’ history of past COVID-19, although this data was not available for all patients since some were diagnosed using antigen testing at home. Therefore, we administered a questionnaire where patients self-reported if they had previously contracted the virus.

Exclusion criteria included pregnancy status, other concomitant biological therapies, other known associated autoimmune rheumatic diseases, treatment with rituximab, HIV, HBV, or HCV disease, and comorbidity with known immunodeficiencies. The study was conducted in accordance with the guidelines of the Declaration of Helsinki. The study was reviewed and approved by the Ethics Committee of the Hospital Clínico San Carlos. Written informed consent was obtained from all individual participants included in the study.

### 2.2. Evaluation of SARS-CoV-2 Humoral Response

Serum samples were analyzed for the detection of anti–SARS-CoV2 antibodies at the Microbiology Department at Hospital Clínico San Carlos. Antibody titers were measured using the SARS-CoV-2 IgG II Quant assay (Abbott Diagnostics, Madrid, Spain) on the Alinity i equipment. The SARS-CoV-2 IgG II Quant Assay is a chemiluminescent microparticle immunoassay (CMIA) used for the qualitative and quantitative determination of IgG antibodies to SARS-CoV-2 in human serum and plasma. This assay is used to monitor the antibody response derived from infection and vaccination against SARS-CoV-2 by determining quantitative IgG titers against the SARS-CoV-2 receptor-binding domain (RBD). The results were expressed as arbitrary units (AU) per milliliter. The positive threshold was 50 AU/mL, following the manufacturer’s recommendation. According to the EP34 Guide of CLSI (21), the ranges of results values that can be reported are 21.0–40,000 AU/mL (analytical measurement range) and 40,000–80,000 AU/mL (extended measurement range).

### 2.3. Evaluation of SARS-CoV-2 Cellular Response

T cell response to SARS-CoV-2 was measured using an IFN-γ ELISA kit (Euroimmun, Lübeck, Germany) within 16-h of blood withdrawal and analyzed on a Triturus analyzer (Grifols S.A., Barcelona, Spain). Human lithium-heparin plasma, obtained after stimulation using the SARS-CoV-2 IGRA stimulation tube set (Euroimmun, Lübeck, Germany), was diluted 1:5 in the sample buffer. Afterward, 100 μL of each calibrator (0.1–400 mUI/mL), controls, and diluted samples were added to high-binding 96 well ELISA plates pre-coated with monoclonal anti-IFN-γ antibodies. After 2 h of incubation at room temperature (RT), plates were washed five times with 350 μL of wash buffer. Subsequently, 100 μL of biotin-labeled anti-interferon-gamma antibody was added to each of the microplate wells and incubated for 30 min at RT. After the following washes as described above, 100 μL peroxidase-labeled streptavidin was added and incubated for 30 min at RT. After five additional washes with wash buffer, 100 μL of 3,3′,5,5′-tetramethylbenzidine/peroxide (TMB/H_2_O_2_) was added to each well, incubating it for 20 min, and the absorbance was read at 450 nm after 30 min of adding the stop solution (sulphuric acid). The interpretation of SARS-CoV-2 IFN-γ antibody testing was as follows: <100 mUI/mL = negative, ≥100 to <200 = borderline, ≥200 = positive. The results were obtained by subtracting the blank value from the values for the specific and non-specific stimulations. Validation criteria were performed according to the manufacturer’s guidelines.

### 2.4. Statistical Analysis

Microsoft Excel (v.14.1.0, Washington, DC, USA) was used for data collection and descriptive data analysis. R software (version 4.0.4, New Jersey, USA) was used for descriptive and statistical data analysis. Categorical variables were compared using Fisher’s exact test or the chi-squared test, as appropriate. Quantitative data were analyzed with the Kruskal-Wallis test or Mann–Whitney U test, as convenient. Values were expressed as means ± standard deviation (SD) or median (IQR), and p-values of less than 0.05 were considered significant. GraphPad Prism software (V.9.1.0, New York, USA) was used to make the violin plots and correlations. We excluded treatment groups with a sample size of ≤2 from our analysis and figures due to the lack of significant data resulting from the small sample size.

## 3. Results

### 3.1. Epidemiological and Immunological Characteristics of the Study Population

A total of 79 patients with IMRDs (mean age 58.2 ± 11.4 years, 60.8% women) and 31 patients with HC (mean age 50.9 ± 13.1 years, 67.7% women) were included in this study. Baseline characteristics of all patients and HC are shown in Table 1. Most patients with IMRDs had rheumatoid arthritis (n = 43, 54.4%), followed by psoriatic arthritis (n = 10, 12.7%), ankylosing spondylitis (n = 14, 17.7%), undifferentiated spondyloarthritis (n = 10, 12.7%), and IBD-associated spondyloarthritis (n = 2, 2.5%). Thirty-two (40.5%) IMRD patients were receiving glucocorticoids (a mean dose of 5.3 mg). Patients were most frequently treated with methotrexate (n = 25, 31.6%), followed by etanercept (n = 18, 22.8%). In addition, 63.3% of patients were receiving a combination of immunosuppressants.

Sixty-three (79.7%) patients received the BNT162b2 vaccine and 16 (20.3%) the mRNA-1273 vaccine. Out of 79 patients, 14 (17.7%) had an anamnesis positive for past SARS-CoV-2 infection confirmed serologically by PCR or antigenic test, similar to HC (16.1% reported previous SARS-CoV-2 infection). Fifty-one (64.6%) patients had no adverse reactions after vaccination.

Among twenty-eight patients (35.4%) with adverse reactions after booster, they had mild and transient symptoms (most frequently 39.3% headache, 32.1% malaise, 28.6% pain in the site of injection, and 7.1% myalgias). No severe adverse reactions have been observed in vaccinated patients up to now. Most patients with IMRDs did not have an outbreak of the rheumatic disease; however, 10.1% (8 of 79 patients) did, and all of them had rheumatoid arthritis.

### 3.2. Humoral Immune Responses to COVID-19 Vaccination

Antibodies against the SARS-CoV-2 RBD of the S protein were analyzed after the third dose of BNT162b2 or the mRNA-1273 vaccine. A detectable anti-RBD antibody response was observed in all HC and in all IMRD patients (100%). However, the magnitude of the HC response (median 18,048 [IQR 10,554–25,664] AU/mL) was significantly higher than that of IMRDs patients (median 6590.3 IQR [2061.5–16,848.3] AU/mL) (*p* < 0.001) (Figure 1A). No differences were identified when patients and controls were stratified by age, sex, and type of vaccine received. Specific anti-SARS-CoV-2 IgG levels were higher in IMRD patients with a previous history of COVID-19, with significant differences (*p* < 0.001) (Figure 1B). No differences were observed in the levels of anti-S1 IgG antibodies when we looked at the different IMRD diseases, but when we compared the humoral response of each IMRD disease with HC, rheumatoid arthritis and ankylosing spondylitis had lower titer levels (*p* < 0.01 and *p* < 0.05, respectively) (Figure 1C). Treatment with methotrexate (MTX) led to a lower titer compared to HC and other IMRD patients not taking any conventional synthetic disease-modifying antirheumatic drugs (csDMARDs) (*p* < 0.01) (Figure 1D). Compared to the HC and no treatment groups, bDMARDs did not significantly decrease antibody titers, with the exception of adalimumab (*p* < 0.01) (Figure 1E). The serologic results were reported as median because, after exploring the dataset using a quantile-quantile plot, the data showed a non-normal distribution.

### 3.3. Cellular Immune Responses to COVID-19 Vaccination

Positive specific cellular responses were displayed in 69 out of 79 (87.4%) patients after the third dose of the SARS-CoV-2 vaccine in our IMRD patients (82.3% when considered IMRDs without previous COVID-19). Median (IQR) IFN-γ levels in IMRDs were 1606.8 (607.1–2056.2) mUI/mL, and in all (100%) HC, median (IQR) IFN-γ levels were 2216 (2015.2–2372.1) mUI/mL, with significant differences between both groups (*p*  < 0.0001) (Figure 2A). Women showed significantly higher responses than men (*p* = 0.04). No differences were identified when patients and controls were stratified by age and type of vaccine received.

Past COVID-19 infection was associated with higher titers than in patients without prior infection (Figure 2B). All patients with prior COVID-19 had positive cellular responses. No differences were observed in the levels of IFN-γ when we compared the different IMRD diseases. However, significant differences were seen when we compared IFN-γ titers of each IMRD disease with HC; rheumatoid arthritis, psoriatic arthritis, and ankylosing spondylitis had lower titer levels (*p*  < 0.0001, *p* < 0.01, and *p* < 0.05, respectively) (Figure 2C). Treatment with MTX and leflunomide led to lower IFN-γ titers compared to HC and other IMRD patients not taking any csDMARDSs (*p*  < 0.0001 and *p* < 0.01, respectively) (Figure 2D). Among bDMARDs, adalimumab, etanercept, and guselkumab appeared to decrease IFN-γ titers compared to HCs and the no treatment group. (*p*  < 0.0001, *p* < 0.001, and *p* < 0.05, respectively) (Figure 2E).

We also observed differences between IMRD patients who had a positive cellular response (responders) and those who did not (non-responders). They are shown in Table 2. Significant differences were seen when stratifying by sex; most of the responders’ patients were female (65.2%, *p* = 0.04). We did not identify significant differences between groups in relation to age, diagnosis, comorbidity, or type of vaccine. In regard to treatment, we did not observe significant differences in most of the csDMARDs, except in the leflunomide group, where half of the non-responders were on leflunomide, compared with 8.5% of the responders who were on it (*p* = 0.0036). Nearly all bDMARDs show no significant differences in cellular response, except in the sarilumab group (*p* = 0.0407). No differences were observed in the tsDMARDs group. All of the IMRD patients that had a prior SARS-CoV-2 infection (hybrid group) had a positive cellular response.

We also focused on the correlation between the two arms of the adaptive immune response. A positive, weak correlation between humoral and cellular immune responses after the third dose of vaccine has been observed in IMRD patients (r = 0.5, *p* < 0.0001) (Figure 3).

### 3.4. SARS-CoV-2 Infection Follow-Up

Our study population was followed up 5 months after the cellular and humoral assay to evaluate whether high cellular IFN-γ responses correlated with protection against subsequent exposure to SARS-CoV-2 infection. Seventeen out of 79 (21.5%) IMRD patients, four of them with negative cellular responses, were further exposed to SARS-CoV-2 after the booster, all of them related to mild symptoms. Eleven out of 31 (35.5%) HCs referred to exposure to SARS-CoV-2. There were no serious or major adverse events in the control group. None of them required hospitalization and are currently free of the disease’s sequelae.

## 4. Discussion

To the best of our knowledge, this is one of the very first studies assessing the immunogenicity of SARS-CoV-2 booster vaccination in patients with IMRDs on immunosuppressive therapy (non-rituximab). We found a detectable, specific SARS-CoV-2 cellular response in 87.4% of IMRD patients (82.3% without previous COVID-19) and in all HC. Specific humoral responses scored positive in all IMRDs and HC. These results are in line with other cohort reports showing that the majority of IMRD patients are able to respond to SARS-CoV-2 vaccines or infection [28,35]. Infection seems to induce higher responses (positive in all naturally infected patients, while only 82.3% were positive in non-infected patients), in line with data published in other populations [36,37], which stresses the need to individualize booster decisions. However, lower titers of IFN-γ and antibodies against the SARS-CoV-2 RBD of the S protein were observed in IMRD patients compared with HC. This reduction in the vaccine’s immunogenicity might be due to immunosuppressive/immunomodulatory therapies [15].

Regarding humoral response after the third vaccine dose (booster), our findings show a substantial reduction with methotrexate and adalimumab treatment [38]. To avoid factor rheumatoid interference in rheumatoid arthritis patients, we only measured IgG antibodies against SARS-CoV-2, which did not significantly interfere, as shown by Liu et al. [39].

Spike-specific T-cell response was also evaluated, displaying significantly lower titers in those patients under methotrexate and leflunomide and the bDMARDs adalimumab, etanercept, and guselkumab. These reductions in cellular and humoral responses are in agreement with other studies [14,16,40]. MTX constitutes a cornerstone treatment for a range of rheumatic diseases. It has been reported that the disruption of MTX for 2 weeks after vaccination improved the adaptive response to influenza vaccination in patients with rheumatoid arthritis with a low risk of disease flares [41]. Moreover, certain biological therapies would enable higher specific vaccine responses with respect to csDMARDs and tsDMARDs. More studies are certainly needed to address the impact of comedication on cellular and humoral immune responses after boosters of SARS-CoV-2 vaccines.

From the viewpoint of particular IMRD at risk of low immunogenic response to SARS-CoV-2 booster, rheumatoid arthritis, psoriatic arthritis, and ankylosing spondylitis were associated with a diminished cellular response to the vaccine compared to healthy controls. These results seem to be at least partially explained by the underlying treatment. Our findings in IMRD patients are in agreement with a study by Mauro et al. [14].

As manifested in our cohort, although the different SARS-CoV-2 vaccines showed a good immunogenicity outline according to adaptive response, a considerable portion of patients with IRMD (12.6%) failed to mount an appropriate cellular response, as shown in other cohorts [26,36]. These non-responders in our cohort were essentially men (n = 7, 8.9%), which might be explained by sexual dimorphism, which has potential implications for the severity and mortality of COVID-19 [42,43]. Another potential explanation of the loss of cellular response might be the occurrence of anti-IFN-γ antibodies described by Bastard et al. [44], which were not performed in our study. The lower immunogenicity of SARS-CoV-2 vaccination in IMRD patients after booster might be due to the development of exhausted lymphocytes, which were not studied [45]. The persistent inflammation seen in patients with rheumatoid arthritis triggers the development of T cells with an exhausted phenotype, which is characterized by a diminished ability to respond to viral antigens [46]. Persistent antigen stimulation with vaccination might increase these phenomena in IMRD patients.

In the case of the hybrid subgroup (patients with IMRD who were naturally immunized by infection plus vaccination), they presented the highest specific anti-SARS-CoV-2 IFN-γ levels, which might indicate that the cellular response induced by prior natural infection was significantly enhanced by subsequent vaccination [47].

The poor correlation between cellular and humoral immunity might be explained by chronic suppression therapy in patients with rheumatological diseases [48]. Only eight patients (10.1%) of our cohort had a flare-up (without requiring medication changes) identified from the post-vaccination surveillance; all of them had rheumatoid arthritis. Similar results were seen in other cohorts studied [49,50]. In contrast, Geisen et al. and Braun-Moscovici et al. reported no flare at all [51,52]. These data should be interpreted with prudence due to an exploratory analysis of disease activity assessment performed within a variable prevaccination and postvaccination time frame.

This study has several limitations that should be mentioned, and thus the results should be interpreted with caution. Our results cannot be considered definitive due to the small sample size of both patients and healthy controls. While our findings do provide a preliminary understanding of the immunogenic response in this subgroup of patients, further studies with larger participant numbers are required to validate our findings. Additionally, the IGRA test does not differentiate between CD4^+^ and CD8^+^ SARS-CoV-2-specific T cells. More research is necessary to gain a deeper understanding of the extent to which each cell subset contributes to the immune response following infection or vaccination. Another limitation is that it was a single-center study, which might limit the impact of the study. Therefore, we present the work as a pilot exploratory analysis.

On the other hand, our study has several strengths. We have recruited homogeneous study groups with patients and HC matched by age and gender. Moreover, we included in the study patients with different IMRDs and treatments and determined the influence of the booster SARS-CoV-2 vaccine on adaptive immune responses. Moreover, the assays used in the present study to detect a SARS-CoV-2 specific response are easy and highly reproducible.

To summarize, our results show that most of our IMRD patients develop cellular and humoral responses after the third dose of the SARS-CoV-2 vaccine. The results displayed have important implications for the management of anti-SARS-CoV-2 vaccination in patients treated with biological therapies. Based on our data, we believe in the necessity of these functional immunological studies to better define the vaccination strategies for individual IMRD patients.

## Figures and Tables

**Figure 1 biomedicines-11-02418-f001:**
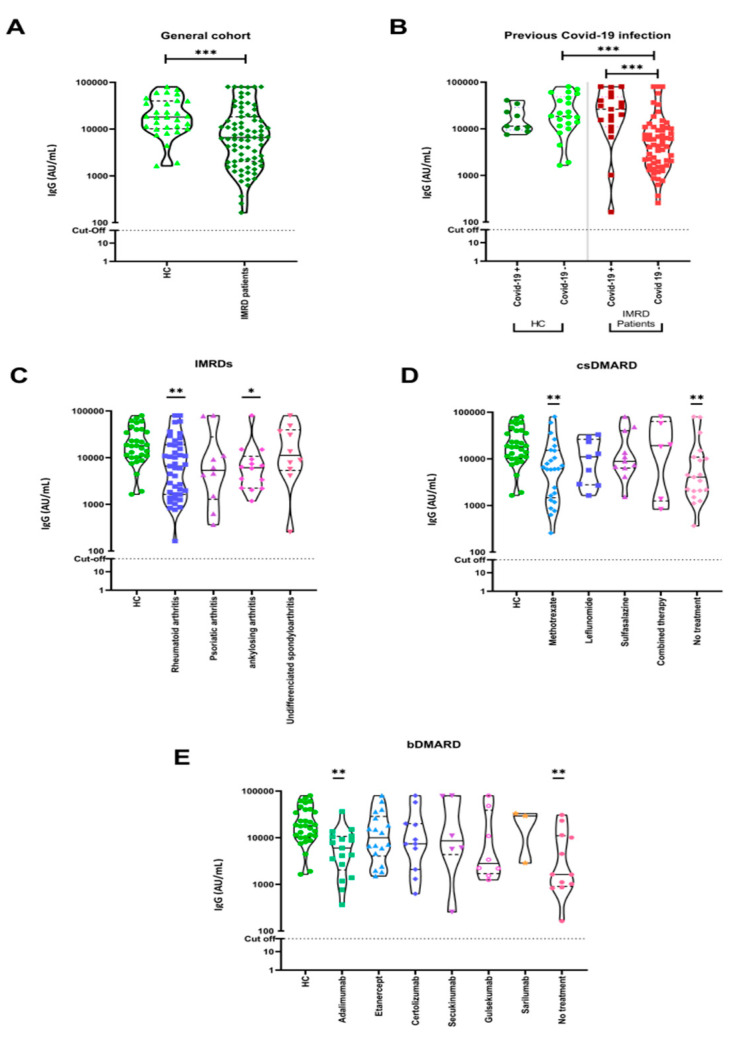
Anti-S1 IgG antibodies in IMRD patients and HC measured by chemiluminescent microparticle immunoassay are represented by a violin plot. Dotted lines represent the positivity cutoff: ≥50 AU/mL. (**A**) Significant differences were observed between our cohort of IMRD patients and HC. (**B**) Antibody titers with and without past COVID-19. (**C**) Levels of anti-SARS-CoV-2 antibodies are compared across different IMRD patients and HC. (**D**) Antibody levels in patients treated with csDMARDs (**E**) and bDMARDs. No treatment group represents IMRD patients without csDMARDs treatment (**D**) or without bDMARDs treatment (**E**). Data were analyzed by Mann–Whitney U or Kruskal-Wallis tests when we compared two or more groups, respectively. * *p* < 0.05, ** *p* < 0.01, *** *p* < 0.001.

**Figure 2 biomedicines-11-02418-f002:**
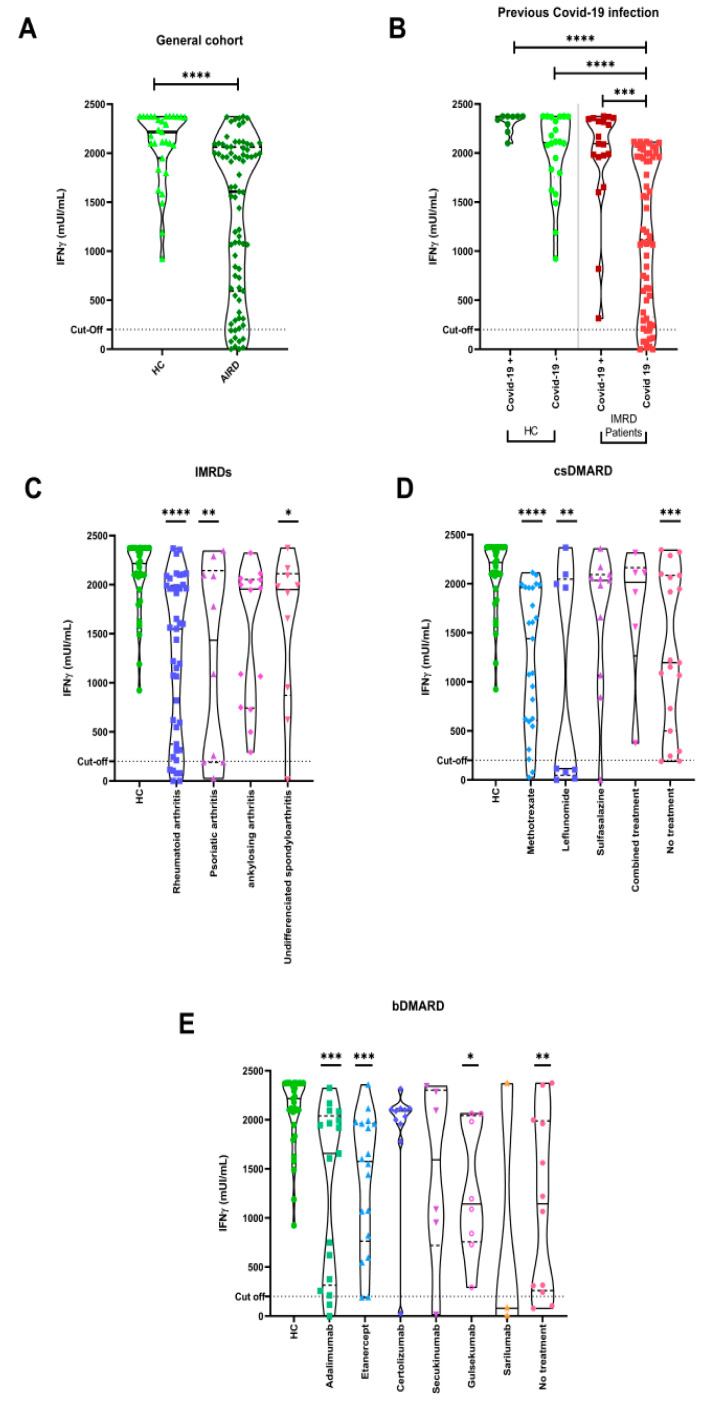
Specific anti-SARS-CoV-2 IFN-γ responses were measured by IGRA. Dotted lines represent the positivity cutoff: ≥200 mUI/mL. (**A**) Significant differences were observed between our cohort of IMRD patients and HC. (**B**) IFN-γ titers with and without past COVID-19. (**C**) Levels of anti-SARS-CoV-2 IFN-γ are compared across different IMRDs and HC. (**D**) IFN-γ levels in patients treated with csDMARDs (**E**) and bDMARDs. Data were analyzed by Mann–Whitney U or Kruskal-Wallis tests when we compared two or more groups, respectively. * *p* < 0.05, ** *p* < 0.01, *** *p* < 0.001, **** *p* < 0.0001.

**Figure 3 biomedicines-11-02418-f003:**
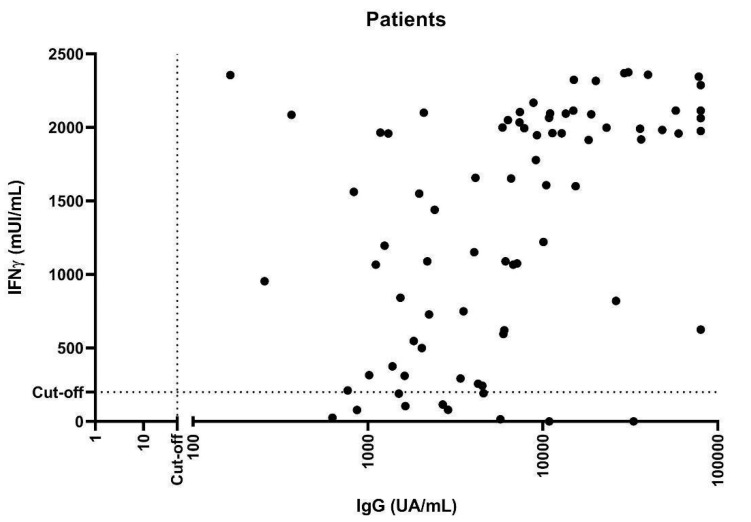
Correlation of anti-S IgG antibody titers and IFN-γ levels after third-dose vaccination. (r = 0.5, *p* < 0.0001).

**Table 1 biomedicines-11-02418-t001:** Demographic and clinical characteristics of patients and controls.

		Healthy Controls (n = 31)	IMRDs Patients (n = 79)
**Age**	Mean ± SD years	50.9 ± 13.1	58.2 ± 11.4
**Sex** n (%)	Female	21 (60.8)	48 (67.7)
	Male	10 (39.2)	31 (32.2)
**Diagnosis** n (%)	Rheumatoid arthritis	N/A	43 (54.4)
	Psoriatic arthritis	N/A	10 (12.7)
	Ankylosing spondylitis	N/A	14 (17.7)
	Undifferentiated spondyloarthritis	N/A	10 (12.7)
	IBD-associated spondyloarthritis	N/A	2 (2.5)
**Medication** n (%)	Glucocorticoids	N/A	32 (40.5)
	Dose, mean ± SD (mg)	N/A	5.3 ±1.7
	**Conventional synthetic DMARDs**		
	Methotrexate	N/A	25 (31.6)
	Sulfasalazine	N/A	11 (13.9)
	Leflunomide	N/A	9 (11.4)
	Mycophenolate	N/A	1 (1.3)
	Hydroxychloroquine	N/A	2 (2.5)
	**Biologicals DMARDs**		
	Etanercept	N/A	18 (22.8)
	Adalimumab	N/A	17 (21.5)
	Certolizumab	N/A	11 (13.9)
	Gulsekumab	N/A	8 (10.1)
	Secukinumab	N/A	6 (7.6)
	Sarilumab	N/A	3 (3.8)
	Infliximab	N/A	2 (2.5)
	Ixekizumab	N/A	1 (1.3)
	Abatacept	N/A	1 (1.3)
	**Targeted Synthetic DMARDs**		
	Baricitinib	N/A	5(6.3)
	Upadacitinib	N/A	4(5.1)
	Combination of conventional and biological DMARDs	N/A	50(63.3)
**3rd COVID-19 vaccine** n (%)	BNT162b2 (Pfizer)	5 (16.1)	63 (79.7)
	mRNA-1273 (Moderna)	26 (83.9)	16 (20.3)
**Prior history** **of COVID-19** n (%)	PCR or IgG antigenic test	5 (16.1)	14 (17.7)

**Table 2 biomedicines-11-02418-t002:** Demographic characteristics of IMRD patients: cellular responders and non-responders.

	Responders	Non-Responders	*p*-Value
	(n = 69) (%)	(n = 10) (%)	
**Age (Median)**	58	62	0.4433
**Female**	45 (65.2)	3 (30.0)	0.0429
**Diagnosis**			0.2672
Rheumatoid arthritis	37 (53.6)	6 (60)	0.7481
Psoriatic arthritis	7 (10.1)	3 (30)	0.1094
Ankylosing spondylitis	14 (20.3)	0	0
Undifferentiated spondyloarthritis	9 (13.1)	1 (10)	1
IBD-associated spondyloarthritis	2 (2.9)	0	0
**Comorbidity**			
Arterial hypertension	25 (36.2)	3 (30)	1
Diabetes mellitus	5 (7.2)	1 (10)	0.5687
Dyslipidemia	27 (39.1)	2 (20)	0.31
Cardiovascular disease	0	1 (10)	0.1266
COPD	3 (4.3)	1 (10)	0.4246
Interstitial lung disease	0	0	-
Oncologic disease	0	0	-
**Treatment**			
Glucocorticoids (mean) (mg/day)	1.79	3.35	0.2171
**Conventional synthetic DMARDs**			0.1249
Methotrexate	31 (45)	2 (20)	0.18
Leflunomide	6 (8.7)	5 (50)	0.0036
Sulfasalazine	12 (17.4)	1 (10)	1
Mycophenolate	1 (1.4)	0	1
Hydroxychloroquine	10 (14.5)	0	1
**Biologicals DMARDs**			0.4556
Adalimumab	15 (21.7)	2 (20)	1
Infliximab	8 (11.6)	0	0.5865
Etanercept	2 (2.9)	0	1
Certolizumab	16 (23.2)	2 (20)	1
Secukinumab	10 (14.5)	1 (10)	1
Ixekizumab	5 (7.24)	1 (10)	0.5687
Gulsekumab	1 (1.4)	0	1
Abatacept	1 (1.4)	0	1
Sarilumab	1 (1.4)	2 (20)	0.0407
**Targeted synthetic DMARDs**			
Baricitinib	4 (5.8)	1 (10)	0.5013
Upadacitinib	4 (5.8)	0	1
**Prior history of COVID-19**	19 (27.5)	0	0.1069
**3rd COVID-19 vaccine**			1
Pfizer	55 (79.7)	8 (80)	
Moderna	14 (20.8)	2 (20)	

## Data Availability

The datasets generated and analyzed during the current study are available from the corresponding author on reasonable request.

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
