# Peer review of "Specific Cellular and Humoral Response after the Third Dose of Anti-SARS-CoV-2 RNA Vaccine in Patients with Immune-Mediated Rheumatic Diseases on Immunosuppressive Therapy"

_biomedicines, 2023, doi:10.3390/biomedicines11092418_

Round 1
Reviewer 1 Report
In their study, Mohamed et al. investigated specific cellular and humoral responses after the third dose of SARS-CoV-2 RNA vaccination in patients with immune-mediated rheumatic diseases (IMRD) receiving biologic therapies. In general, studies investigating the antigenicity of different SARS-CoV-2 vaccination regimens in individuals belonging to vulnerable groups such as the elderly or patients with immune disorders have important implications for specific vaccination schedules or personalized decision essential to obtain proper SARS-CoV-2 specific responses in these individuals. Thus, the current study represents an important contribution to the field.
Overall, the study is well written and the methods are adequately described. However, a few points should be clarified in order to draw clear conclusions.
First, the labeling of all graphs on both axes is too small and very difficult to read. This should be improved.
Biologic therapies should be explained to readers not familiar with the field.
The authors mentioned that 40.5% of IMRD patients received glucocorticoids. Glucocorticoids have been shown to influence the response to SARS-CoV-2 vaccination. It would be worthwhile to include the effects of glucocorticoids in the current study, at least by distinguishing patients with and without glucocorticoids in the different treatment groups by different symbols.
Furthermore, it is not clear what the "no treatment" group in the figures (D and E) means for both humoral and cellular responses, especially since this group is not identical between D and E compared to HC values. Furthermore, if "no treatment" means that a group of IMRD patients did not receive any biological therapies, then it would be implied that IMRD itself leads to reduced responses to SARS-CoV-2 boost vaccination. In this case, it would mean that certain biologic therapies would actually improve vaccine response in IMRD patients. This point needs to be clarified.
Author Response
Specific cellular and humoral response after the third dose of anti-SARS-CoV-2 RNA vaccine in patients with IMRDs on biologic therapy
Dear reviewers and editors,
Thank you for taking the time to review our manuscript thoroughly and provide insightful feedback. We have carefully revised it and incorporated each of your comments, which are marked in yellow throughout the document.
Editor
1) Authors should report in the discussion the fact that the present study comprises few patients in each group of medication, and further study would be necessary to strengthen their conclusions.
Thank you very much for your comment. We agree we cannot draw definitive conclusions with our current study due to the small cohort size. We have pointed out this in the discussion as a limitation of our work and that our results warrant further studies with a larger number of participants to validate our conclusions.
Reviewer 1
1) First, the labeling of all graphs on both axes is too small and very difficult to read. This should be improved.
Thank you for this comment. We have made the changes suggested in the Results section.
2) Biologic therapies should be explained to readers not familiar with the field.
Thank you for this relevant point. We have made the necessary correction by replacing "biological therapies" with "immunosuppressive therapy". This category includes conventional synthetic disease-modifying antirheumatic drugs (csDMARDs), biological disease-modifying antirheumatic drugs (bDMARDs), and targeted synthetic disease-modifying antirheumatic drugs (tsDMARDs). A short summary of their details has been included in the introduction. Tables 1 and 2 list every drug in each respective category. By doing so, readers will have a clear understanding of the specific immunosuppressive therapies mentioned in the article and tables.
3) The authors mentioned that 40.5% of IMRD patients received glucocorticoids. Glucocorticoids have been shown to influence the response to SARS-CoV-2 vaccination. It would be worthwhile to include the effects of glucocorticoids in the current study, at least by distinguishing patients with and without glucocorticoids in the different treatment groups by different symbols.
We appreciate your essential appreciation, since we agree that corticosteroids have been identified as a risk factor for the low immunogenicity of vaccines in this subgroup of patients. As you mentioned, 40.5% of the patients in our study received low-dose glucocorticoid treatment (mean dose of 5.3 mg per day of prednisone), which we did not study directly in the first analysis compared with the control group. We did not make a first identification of these patients, which is very interesting for future studies.
Nevertheless, we did analyze them when comparing the group of responders and "non-responders" of the cellular response (Table 2), where we did not observe statistically significant differences in the doses of corticosteroids used in both groups (1.79 mg and 3.35 mg daily of prednisone respectively, with a p= 0.2171) related with the cellular response.
4) It is not clear what the "no treatment" group in the figures (D and E) means for both humoral and cellular responses, especially since this group is not identical between D and E compared to HC values.
Thank you for your comment. We agree that it is crucial to clarify the information in the figures. Although we commented on it in the results, we will make sure to specify it in the figure caption. In both the humoral and cellular responses, figure D “no treatment” group are IMRD patients without conventional synthetic Disease-Modifying Antirheumatic Drugs (csDMARD). In figure E, the "no treatment" group are IMRD patients not taking any biological Disease-Modifying Antirheumatic Drugs (bDMARD).
5) Furthermore, if "no treatment" means that a group of IMRD patients did not receive any biological therapies, then it would be implied that IMRD itself leads to reduced responses to SARS-CoV-2 boost vaccination. In this case, it would mean that certain biologic therapies would actually improve vaccine response in IMRD patients. This point needs to be clarified.
We appreciate your positive feedback and critical point. While we understand that our analysis may have been interpreted in a certain way, we want to clarify that the group of non-biological treatments refers to patients who are not receiving biological immunosuppressive treatment (bDMARDs), but may be receiving other types of immunosuppressants like corticosteroids, csDMARDs or tsDMARDs. We have stated now this point in the revised manuscript. Therefore, we cannot confirm that these patients with “no treatment” were not on other types of immunosuppressants that could potentially lower their immune response. As you mention, this might suggest that certain biologic therapies would enable higher specific vaccine responses with respect to csDMARDs and tsDMARDs. We have added this sentence to the Discussion section.
Furthermore, we have conducted a thorough evaluation of the humoral and cellular response in each IMRD to address the important point you raised about whether the IMRD itself reduces responses to vaccination. Our results showed that while the rheumatoid arthritis and ankylosing spondylitis groups had positive titres in the humoral response, they were lower than the HC group. Similarly, in the cellular response, patients with rheumatoid arthritis, psoriatic arthritis, and ankylosing spondylitis had IFN-γ titers that were positive but lower than the HC group.

Reviewer 2 Report
Manuscript: Specific cellular and humoral response after the third dose of ani-
SARS-CoV-2 RNA vaccine in patients with IMRDs on biologic therapy
Notes:
1. The authors describe a study where patients with immune-mediated rheumatic diseases are provided with a booster SARS CoV-2 RNA vaccine. Both the humoral antibody response and the cellular immune response were evaluated and compared to a control group of healthy individuals. The humoral response was measured by a commercial ELISA assay from Abbott Diagnostics which yields results in arbitrary units. The cellular immune response was assessed by an IGRA based assay from Euroimmun, which stimulated cells in whole blood collected in lithium heparin tubes containing peptides from SARS CoV-2 virus. The resulting stimulation resulted in the production of IFN-g, which was subsequently measured via ELISA. The results were based on assay thresholds expressed in units/ml.
2. The clinical data generated were presented in graph form and tables (demographic data). The graphs were very difficult to read due to extremely small font size. The colors selected do not allow the ready to easily identify the mean values. Recommend increasing the font size to enable the ready to interpret the data. This includes titles, axis labels, and asterisks for the statistical significance notes. Consider box plots in lieu of violin plots to show the data. The cut off values on the graphs are difficult to see, consider using color and bold lines to help visualize the cut off values.
3. The statistical analysis showed interesting correlations that would not be unexpected in these patient populations but as the authors mentioned, not much data has been shown prior to this study,
4. There are some issues with the data that should be addressed.
a. The stimulation of cells in whole blood taken from IMRD subjects in the IGRA tubes may be influenced by therapeutics or anti-inflammatory drugs present in the sample. These would inhibit the cellular immune response in the IGRA assay, likely lowering the measured IFN-g. Thus, reduced IGRA results would be a combination of suppressed immunogenicity in the patient and suppressed response in the assay itself by the drug. The data in Figure 3 may be an example of this. Could the samples that have elevated IgG levels but fall below the cut-off value of the IFN-g assay be caused by immunosuppressive drugs in the sample?
b. A suggestion would be to amend the table(s) of the drugs present (or likely to be present) in subjects with some notes on how IGRA assay could have been impacted. If the authors have data (if available) from validation experiments where the drugs were spiked into the IRGA assays, that should be presented as well.
5. Rheumatoid factor: The authors show data from Rheumatoid arthritis (RA) subjects. As the authors are likely aware, RA subjects are known to have Rheumatoid factor (auto-reactive antibodies) which can interfere with the measurement of IgG in immunoassays (i.e. SARS-CoV2 ELISA) assay. Would the authors be able to comment on this? Were any of the results from RA subjects possibly skewed by the presence of RA factor?
Author Response
1) The clinical data generated were presented in graph form and tables (demographic data). The graphs were very difficult to read due to extremely small font size. The colors selected do not allow the ready to easily identify the mean values. Recommend increasing the font size to enable the ready to interpret the data. This includes titles, axis labels, and asterisks for the statistical significance notes. Consider box plots in lieu of violin plots to show the data. The cut off values on the graphs are difficult to see, consider using color and bold lines to help visualize the cut off values.
Thank you for this comment. We have made the changes suggested in the Results section.
2) The stimulation of cells in whole blood taken from IMRD subjects in the IGRA tubes may be influenced by therapeutics or anti-inflammatory drugs present in the sample. These would inhibit the cellular immune response in the IGRA assay, likely lowering the measured IFN-g. Thus, reduced IGRA results would be a combination of suppressed immunogenicity in the patient and suppressed response in the assay itself by the drug. The data in Figure 3 may be an example of this. Could the samples that have elevated IgG levels but fall below the cut-off value of the IFN-g assay be caused by immunosuppressive drugs in the sample? A suggestion would be to amend the table(s) of the drugs present (or likely to be present) in subjects with some notes on how IGRA assay could have been impacted. If the authors have data (if available) from validation experiments where the drugs were spiked into the IRGA assays, that should be presented as well.
Thank you so much for highlighting these points. We measured the adaptive immune response two weeks after stopping treatment to reduce the influence of anti-inflammatory drugs in the assays. We have added this in the study design section.
3) Rheumatoid factor: The authors show data from Rheumatoid arthritis (RA) subjects. As the authors are likely aware, RA subjects are known to have Rheumatoid factor (auto-reactive antibodies) which can interfere with the measurement of IgG in immunoassays (i.e. SARS-CoV2 ELISA) assay. Would the authors be able to comment on this? Were any of the results from RA subjects possibly skewed by the presence of RA factor?
Thank you for this observation that is worth mentioning. As we know, rheumatoid factors (RF) are commonly IgM antibodies. In the measurement of IgG antibodies against SARS-CoV-2, RF did not significantly interfere as shown by Liu et al. We have added this point in the Discussion section.
Liu W, Long X, Wan K, Yin M, Yin Y, Zhang B, Li L, Song Y. The endogenous factors affecting the detection of serum SARS-CoV-2 IgG/IgM antibodies by ELISA. J Med Virol. 2022 May;94(5):1976-1982. doi: 10.1002/jmv.27557. Epub 2022 Jan 7. PMID: 34967441; PMCID: PMC9015225.

Reviewer 3 Report
The authors are correct in stating that the adaptive immune responses to SARS-CoV-2 vaccination in patient son immunosuppressive therapy such as patients with immune-mediated rheumatic diseases (IMRDs) are still insufficiently studied. Thus, new studies investigating the impact of different medications are valuable. However, the present study comprises too few patients in each group of medication to allow for meaningful conclusions. Stratification with a reduced number of subgroups (e.g. treatment vs no treatment) might allow for more robust data, but may not serve the purpose of the study. In addition to this major shortcoming the study has several other important flaws.
Specific points of criticism:
1. Title:
1. IMRDs should be spelled out.
2. “on biologic therapy”: The study also includes cDMARDs!
2. Abstract:
1. Abbreviations for the different diseases such as “RA” are not necessary in the abstract (no second use in abstract). However, abbreviations such as “IBD” and “bDMARDs” need to be defined.
2. Results: Antibody responses need to be mentioned.
3. Conclusion: Define “adequate specific adaptive response”!
3. Key messages,
1. Mention antibody together with cellular responses!
2. “need for reconsidering these immunological studies”: In which way should the studies be reconsidered? What do the authors suggest regarding the design of these studies?
4. Introduction, 1. paragraph: Which of the parameters does the percentage 7% relate to?
5. Introduction, 2. paragraph: Reference 9 is not an influenza or pneumococcus vaccination study!
6. Introduction, 3. paragraph: Include more references (“studies”)!
7. Introduction, 4. paragraph: “…T-cell immunity is the most effective in viral host immunity.” => This statement is imprecise/not correct. There is strong evidence that the cellular immune response persists longer than the antibody response and is more robust against the different SARS-CoV-2 variants. However, in general neutralizing antibodies are very effective in preventing viral infection of the cells. Provide references!
8. Introduction, 4. paragraph: “…CD8+ T-cell responses may play a protective role in the presence of decreasing or subprotective antibody titers” => CD4+ T-cell responses also play an important role and are in fact more prominent that CD8+ T-cells responses.
9. Introduction, 5. paragraph: “Published data regarding immune response identify drugs targeting B cells as risk factors for low immunogenicity.” => references!
10. Introduction, last paragraph: What is meant by “prophylactic strategies”?
11. Introduction, last paragraph: “identify… as well as the correlate of protection in this population” => The study does not allow for a definition of correlate of protection.
12. Materials and methods, Study design: “questionnaire including questions on age, gender, type of rheumatic diagnosis, pharmacologic history, COVID-19 vaccination status and history of past-COVID-19” => Was the source for the analysis of the demographic and clinical data of the patients these questionnaires or the electronic clinical database? It should be the latter!
13. Materials and methods, Evaluation of SARS-CoV-2 cellular response: It should be mentioned (since the authors give a lot of details of the assay protocol)
1. that the BLANK value is subtracted from the values for the specific and unspecific stimulations.
2. if the validation criteria suggested by the manufacturer were applied.
14. Materials and methods, Statistical analysis: Provide more detailed information:
1. Which software was used for which analysis?
2. Which software/test was used for the correlation analysis the results of which are shown in Fig.3?
15. Results, Figs. 1 and Fig.2:
1. The plots are much too small, axes titles, numbering and labeling are hardly readable.
2. In C, D and E, it is not clear between which groups the statistically significant results were observed.
3. In Table 1, additional treatments are listed, which are not represented in the figures (e.g. Mycophenolate). (How) were these treatments included in the analyses?
16. Results, Table 1:
1. The numbers for female and male HCs and patients are wrong (interchanged)!
2. Prior history of COVID-19: PCR or IgG => add “or antigenic test”
3. Figs. 1 and 2 show data for the “combined therapy” and “no treatment” groups (in D and E); however, this groups are not listed in Tables 1 and 2!?
4. Check formatting (also applies to Table 2)
17. Results, Humoral immune responses to COVID-19 vaccination:
1. According to the manufacturer, results are given in AU/mL, not UA/mL. => correct in main text and in Fig. 1
2. “Specific anti-SARS-CoV-2 IgG levels were higher IMRD patients with a previous history of COVID-19 with significant differences (p<0.001) (Figure 1B).” => This is known to be true also in healthy individuals. Was the difference between HC +COVID-19 and HC- COVID-19 not statistically significant?
3. csDMARDs: define!
4. “Nearly all bDMARDs appeared to influence antibody titers except for adalimumab (p<0.01) (Figure 1E).” => influence in which way?
18. Results, Cellular immune responses to COVID-19 vaccination:
1. “Among bDMARDs, adalimumab, etanercept and guselkumab appeared to influence IFN-γ titers except for adalimumab (p <0.0001, p<0.001 and p<0.05, respectively) (Figure 2E).” => contradiction?!
Moderate editing of English language is required.
Author Response
1) The present study comprises too few patients in each group of medication to allow for meaningful conclusions.
Thank you for your comment. We agree with you, and so does the editor, that due to the limited number of patients according to different therapies and healthy controls studied, we cannot definitively draw conclusions based on our preliminary findings. As commented above for Reviewer #1, we have stated this point as a limitation of the study. We have added that it is necessary to conduct further studies with larger sample sizes to confirm our results. In the discussion, we have pointed out this limitation.
2) Title:
1. IMRDs should be spelled out.
2. “on biologic therapy”: The study also includes cDMARDs
Thank you for your suggestions. We have implemented the necessary changes accordingly.
3) Abstract:
1. Abbreviations for the different diseases such as “RA” are not necessary in the abstract (no second use in abstract). However, abbreviations such as “IBD” and “bDMARDs” need to be defined.
2. Results: Antibody responses need to be mentioned.
3. Conclusion: Define “adequate specific adaptive response”
Thank you for the suggested corrections. We have implemented them and the abstract is now more concise and easier to understand.
4) Key messages:
1. Mention antibody together with cellular responses!
2. “need for reconsidering these immunological studies”: In which way should the studies be reconsidered? What do the authors suggest regarding the design of these studies?
Thank you for your appreciation, we have rephrased the sentence as follows "Our data underline the need to take these immunological studies into account when deciding on the need for additional boosters on a personalized basis". By this, we mean that studies of the cellular and humoral responses in this subgroup of patients can help clinicians to decide the need for revaccination according to the immune response of each patient.
4. Introduction, 1. paragraph: Which of the parameters does the percentage 7% relate to?
Thank you for highlighting this. The 7% was an error, it does not correspond to any of the above and we have removed it.
- Introduction, 2. paragraph: Reference 9 is not an influenza or pneumococcus vaccination study!
Thank you for pointing out the mistake. Reference 9 actually corresponds to the previous paragraph's references. We have made the necessary correction.
- Introduction, 3. paragraph: Include more references (“studies”)!
Thank you for your appreciation. We have referenced the previous studies conducted, which we have also cited in the discussion:
- Furer, V.; Eviatar, T.; Zisman, D.; Peleg, H.; Paran, D.; Levartovsky, D.; Zisapel, M.; Elalouf, O.; Kaufman, I.; Meidan, R.; et al. Immunogenicity and Safety of the BNT162b2 MRNA COVID-19 Vaccine in Adult Patients with Autoimmune Inflammatory Rheumatic Diseases and in the General Population: A Multicentre Study. Ann Rheum Dis 2021, 80, 1330–1338.
- Boekel, L.; Steenhuis, M.; Hooijberg, F.; Besten, Y.R.; van Kempen, Z.L.E.; Kummer, L.Y.; van Dam, K.P.J.; Stalman, E.W.; Vogelzang, E.H.; Cristianawati, O.; et al. Antibody Development after COVID-19 Vaccination in Patients with Autoimmune Diseases in the Netherlands: A Substudy of Data from Two Prospective Cohort Studies. Lancet Rheumatol 2021, 3, e778–e788.
- Zheng, Y.Q.; Li, H.J.; Chen, L.; Lin, S.P. Immunogenicity of Inactivated COVID-19 Vaccine in Patients with Autoimmune Inflammatory Rheumatic Diseases. Sci Rep 2022, 12, doi:10.1038/s41598-022-22839-0.
- Raptis, C.E.; Berger, C.T.; Ciurea, A.; Andrey, D.O.; Polysopoulos, C.; Lescuyer, P.; Maletic, T.; Riek, M.; Scherer, A.; von Loga, I.; et al. Type of MRNA COVID-19 Vaccine and Immunomodulatory Treatment Influence Humoral Immunogenicity in Patients with Inflammatory Rheumatic Diseases. Front Immunol 2022, 13, doi:10.3389/fimmu.2022.1016927
- Sugihara, K.; Wakiya, R.; Shimada, H.; Kameda, T.; Nakashima, S.; Kato, M.; Miyagi, T.; Mizusaki, M.; Mino, R.; Nomura, Y.; et al. Immunogenicity against the BNT162b2 MRNA COVID-19 Vaccine in Rheumatic Disease Patients Receiving Immunosuppressive Therapy. Internal Medicine 2022, 61, 1953–1958, doi:10.2169/internalmedicine.9223-21.
- Pri-Paz Basson, Y.; Tayer-Shifman, O.E.; Naser, R.; Tartakover Matalon, S.; Kimhi, O.; Gepstein, R.; Halperin, T.; Ziv-Baran, T.; Ziv, A.; Parikh, R.; et al. Immunogenicity and Safety of the MRNA-Based BNT162b2 Vaccine in Systemic Autoimmune Rheumatic Diseases Patients. Clin Rheumatol 2022, 41, 3879–3885, doi:10.1007/s10067-022-06348-z.
- Introduction, 4. paragraph: “…T-cell immunity is the most effective in viral host immunity.” => This statement is imprecise/not correct. There is strong evidence that the cellular immune response persists longer than the antibody response and is more robust against the different SARS-CoV-2 variants. However, in general neutralizing antibodies are very effective in preventing viral infection of the cells. Provide references!
Thank you for pointing this out. As you mentioned, humoral response is an essential immunoprotective barrier against reinfection. However, resolution of viral infection is reliant more on cellular response, as many viruses spread directly between cells without reentering the extracellular environment where antibodies are present. This is supported by patients with agammaglobulinemia or predominantly antibody deficiencies, in which just few viruses’ families are more prevalent, such as herpesviruses and enteroviruses.
Mueller SN, Rouse BT. Immune responses to viruses. Clinical Immunology. 2008:421–31. doi: 10.1016/B978-0-323-04404-2.10027-2. Epub 2009 May 15. PMCID: PMC7151814.
- Introduction, 4. paragraph: “…CD8+ T-cell responses may play a protective role in the presence of decreasing or subprotective antibody titers” => CD4+ T-cell responses also play an important role and are in fact more prominent that CD8+ T-cells responses.
Thank you for this comment. We totally agree with you with this statement. What we want to expose in this sentence is that in a deficiency of humoral immunity, CD8+ T cell response is essential as a defense against viruses as specific cytotoxic cells.
- Introduction, 5. paragraph: “Published data regarding immune response identify drugs targeting B cells as risk factors for low immunogenicity.” => references!
Thank you for bringing this to our attention. We have included references to the primary studies that demonstrate the potential risk of low immunogenicity associated with treatments targeting B cells.
- Furer, V.; Eviatar, T.; Zisman, D.; Peleg, H.; Paran, D.; Levartovsky, D.; Zisapel, M.; Elalouf, O.; Kaufman, I.; Meidan, R.; et al. Immunogenicity and Safety of the BNT162b2 MRNA COVID-19 Vaccine in Adult Patients with Autoimmune Inflammatory Rheumatic Diseases and in the General Population: A Multicentre Study. Ann Rheum Dis 2021, 80, 1330–1338.
- Zheng, Y.Q.; Li, H.J.; Chen, L.; Lin, S.P. Immunogenicity of Inactivated COVID-19 Vaccine in Patients with Autoimmune Inflammatory Rheumatic Diseases. Sci Rep 2022, 12, doi:10.1038/s41598-022-22839-0.
- Moor, M.B.; Suter-Riniker, F.; Horn, M.P.; Aeberli, D.; Amsler, J.; Möller, B.; Njue, L.M.; Medri, C.; Angelillo-Scherrer, A.; Borradori, L.; et al. Humoral and Cellular Responses to MRNA Vaccines against SARS-CoV-2 in Patients with a History of CD20 B-Cell-Depleting Therapy (RituxiVac): An Investigator-Initiated, Single-Centre, Open-Label Study. Lancet Rheumatol 2021, 3, e789–e797.
- Mrak, D.; Simader, E.; Sieghart, D.; Mandl, P.; Radner, H.; Perkmann, T.; Haslacher, H.; Mayer, M.; Koblischke, M.; Hofer, P.; et al. Immunogenicity and Safety of a Fourth COVID-19 Vaccination in Rituximab-Treated Patients: An Open-Label Extension Study. Ann Rheum Dis 2022, 81, 1750–1756, doi:10.1136/annrheumdis-2022-222579.
- Schumacher, F.; Mrdenovic, N.; Scheicht, D.; Pons-Kühnemann, J.; Scheibelhut, C.; Strunk, J. Humoral Immunogenicity of COVID-19 Vaccines in Patients with Inflammatory Rheumatic Diseases under Treatment with Rituximab: A Case-Control Study (COVID-19VacRTX). Rheumatology (United Kingdom) 2022, 61, 3912–3918, doi:10.1093/rheumatology/keac036.
- Introduction, Last paragraph: What is meant by “prophylactic strategies”
Thank you for your appreciation.
We agree that the term "prophylactic strategies" can be confusing. We want to refer to vaccination strategies in these patients.
We have rephrased it as follows: Given the need to implement vaccine strategies in these patients, our aim was to describe the serological and T-cell responses after the third dose of vaccine in a cohort of patients with IMRDs (rheumatoid arthritis and spondyloarthropathies) treated with immunosuppressive therapy (csDMARDs, bDMARDs and tsDMARDs). The aim was to identify the impact of these treatments on vaccine response and to determine which patients would benefit from different vaccination strategies. In addition, we sought to establish correlates of vaccine response and protection in this patient population.
- Introduction, Last paragraph: “identify… as well as the correlate of protection in this population” => The study does not allow for a definition of correlate of protection.
Thank you for your comment. We have followed-up our cohort 5 months after the adaptive response assays to evaluate a potential correlate for protection. This was described in the SARS-CoV-2 Infection Follow-Up section.
- Materials and methods, Study design: “questionnaire including questions on age, gender, type of rheumatic diagnosis, pharmacologic history, COVID-19 vaccination status and history of past-COVID-19” => Was the source for the analysis of the demographic and clinical data of the patients these questionnaires or the electronic clinical database? It should be the latter!
Thank you for your comment. We agree we need to clarify this section.
The data on age, gender, type of rheumatic diagnosis, pharmacologic history, COVID-19 vaccination status were collected from the electronic clinical database. We also checked in the electronic clinical database the history of past-COVID-19, although not all of them were recorded, as the diagnosis of many of them was made by means of antigen testing at home, so we carried out the questionnaire in which it was the patient who reported having passed the infection.
- Materials and methods, Evaluation of SARS-CoV-2 cellular response: It should be mentioned (since the authors give a lot of details of the assay protocol)
1. that the BLANK value is subtracted from the values for the specific and unspecific stimulations.
2. if the validation criteria suggested by the manufacturer were applied.
Thank you for your recommendations. We have added them in the materials and methods section.
14. Materials and methods, Statistical analysis: Provide more detailed information:
1. Which software was used for which analysis?
2. Which software/test was used for the correlation analysis the results of which are shown in Fig.3?
Thank you for your comments. We have provided more information in the Statistical analysis section.
15. Results, Figs. 1 and Fig.2:
1. The plots are much too small, axes titles, numbering and labeling are hardly readable.
2. In C, D and E, it is not clear between which groups the statistically significant results were observed.
Thank you for these constructive comments. We have made the changes suggested in the Results section, as previously mentioned (Reviewer #1).
3. In Table 1, additional treatments are listed, which are not represented in the figures (e.g. Mycophenolate). (How) were these treatments included in the analyses?
Thank you for your comment. We did not include groups of treatments with an n equal to or less than 2 in our figures, as is the case with mycophenolate, because the data would not be significant with such a small sample size.
16. Results, Table 1:
1. The numbers for female and male HCs and patients are wrong (interchanged)!
Thank you for this appreciation. We have corrected this mistake.
2. Prior history of COVID-19: PCR or IgG => add “or antigenic test”
3. Figs. 1 and 2 show data for the “combined therapy” and “no treatment” groups (in D and E); however, this groups are not listed in Tables 1 and 2!?
4. Check formatting (also applies to Table 2)
Thank you for your suggestions. We have added them in the results section.
17. Results, Humoral immune responses to COVID-19 vaccination:
1. According to the manufacturer, results are given in AU/mL, not UA/mL. => correct in main text and in Fig. 1
Thank you for this comment. We have made the changes suggested in the Figure 1 and in main text.
2. “Specific anti-SARS-CoV-2 IgG levels were higher IMRD patients with a previous history of COVID-19 with significant differences (p<0.001) (Figure 1B).” => This is known to be true also in healthy individuals. Was the difference between HC +COVID-19 and HC- COVID-19 not statistically significant?
Thank you for this comment. In our cohort, the difference between infected and uninfected HCs was not statistically significant, probably due to the small sample size.
3. csDMARDs: define!
Thank you for your comment, we have defined the meaning of csDMARDS in the results section.
4. “Nearly all bDMARDs appeared to influence antibody titers except for adalimumab (p<0.01) (Figure 1E).” => influence in which way?
We appreciate your comment and acknowledge that the previous sentence was unclear. Therefore,we have rephrased it as follows: Compared to the HC and no treatment group, bDMARDs did not significantly decrease antibody titres, with the exception of adalimumab.
18. Results, Cellular immune responses to COVID-19 vaccination:
1. “Among bDMARDs, adalimumab, etanercept and guselkumab appeared to influence IFN-γ titers except for adalimumab (p <0.0001, p<0.001 and p<0.05, respectively) (Figure 2E).” => contradiction?!
Thank you for your appreciation. We acknowledge that there was an error, but we have rectified it by making the following corrections: Among bDMARDs, adalimumab, etanercept and guselkumab appeared to decrease IFN-γ titers compared to HCs and the no treatment group.

Reviewer 4 Report
Thank you for the opportunity to review your manuscript entitled "Specific cellular and humoral response after the third dose of anti-SARS-CoV-2 RNA vaccine in patients with IMRDs on biologic therapy". I have carefully evaluated your work and would like to respectfully provide some comments and suggestions for improvement.
1. Please specify in the caption of Figure 1 which data were analyzed using which significance test. This will provide clarity and allow for better understanding of the results.
2. Please provide a full spelling and brief description of csDMARDs and bDMARDs. This will help readers who are not familiar with these abbreviations to understand the text better.
3. Please discuss why there seems to be little correlation between the humoral and cellular immune responses in individual patients, as shown in Figure 3. This will help to provide a more comprehensive understanding of the results and their implications.
Author Response
1. Please specify in the caption of Figure 1 which data were analyzed using which significance test. This will provide clarity and allow for better understanding of the results.
Thank you for your comment. We have clarified it in the caption of Figure 1.
2. Please provide a full spelling and brief description of csDMARDs and bDMARDs. This will help readers who are not familiar with these abbreviations to understand the text better.
Thank you for your comment. We have made some corrections to clarify the immunosuppressive therapies studied for our readers. This category comprises of three types of drugs: conventional synthetic disease-modifying antirheumatic drugs (csDMARDs), biological disease-modifying antirheumatic drugs (bDMARDs), and targeted synthetic disease-modifying antirheumatic drugs (tsDMARDs). A short summary of their details has been included in the introduction. Tables 1 and 2 list every drug in each respective category.
3. Please discuss why there seems to be little correlation between the humoral and cellular immune responses in individual patients, as shown in Figure 3. This will help to provide a more comprehensive understanding of the results and their implications.
Thank you for highlighting this. This poor correlation between cellular and humoral immunity might be explained by the chronic suppression therapy in patients with rheumatological diseases. This point was added to the discussion section.
Oyaert M, De Scheerder MA, Van Herrewege S, Laureys G, Van Assche S, Cambron M, Naesens L, Hoste L, Claes K, Haerynck F, Kerre T, Van Laecke S, Van Biesen W, Jacques P, Verhasselt B, Padalko E. Evaluation of Humoral and Cellular Responses in SARS-CoV-2 mRNA Vaccinated Immunocompromised Patients. Front Immunol. 2022 22;13:858399.

Round 2
Reviewer 1 Report
The author replied all my concerns satisfactorily and improved the quality of the manuscript.
Author Response
Thank you very much for your comment.
Reviewer 3 Report
The majority of the reviewer's comments/points of criticism have been answered in a satisfactory manner. Some points, however, require further revision (see review (2)).

Author Response
Introduction, 4. paragraph: “…T-cell immunity is the most effective in viral host immunity.” => This statement is imprecise/not correct. There is strong evidence that the cellular immune response persists longer than the antibody response and is more robust against the different SARS-CoV-2 variants. However, in general neutralizing antibodies are very effective in preventing viral infection of the cells. Provide references!
Reviewer: Not accepted. Such a general statement cannot be referenced with only one publication. In addition, Mueller and Rouse themselves state at the beginning of their review: “In most situations, defense against viruses involves multiple immune components, and the impact of a single mechanism varies greatly according to the method by which individual viruses enter, replicate, and spread within the host.” AS an example, the following sentence would be more appropriate: “There is a large body of evidence that the SARS-CoV-2-specific T cell response is essential for viral clearance, disease outcome and long-term memory.” Provide references!
Thank you for your recommendations. We have added them in the introduction section.
Almendro-Vázquez P, Laguna-Goya R, Paz-Artal E. Defending against SARS-CoV-2: The T cell perspective. Front Immunol. 2023 Jan 27;14:1107803.
Kared H, Redd AD, Bloch EM, Bonny TS, Sumatoh H, Kairi F, Carbajo D, Abel B, Newell EW, Bettinotti MP, Benner SE, Patel EU, Littlefield K, Laeyendecker O, Shoham S, Sullivan D, Casadevall A, Pekosz A, Nardin A, Fehlings M, Tobian AA, Quinn TC. SARS-CoV-2-specific CD8+ T cell responses in convalescent COVID-19 individuals. J Clin Invest. 2021 Mar 1;131(5):e145476.
Introduction, 4. paragraph: “…CD8+ T-cell responses may play a protective role in the presence of decreasing or subprotective antibody titers” => CD4+ T-cell responses also play an important role and are in fact more prominent that CD8+ T-cells responses.
Thank you for this comment. We provided more references.
- Introduction, Last paragraph: What is meant by “prophylactic strategies”
11.Introduction, Last paragraph: “identify… as well as the correlate of protection in this population” => The study does not allow for a definition of correlate of protection.
Reviewer: Only partially accepted. This study did not establish correlates of vaccine response and protection in this patient population.
Reviewer: Not accepted. Correlates of protection are thresholds (of antibody concentrations, neutralization activity, etc.) above which patients are protected from infection, asymptomatic infection, infection with mild/moderate symptoms, severe disease or death.
Thank you for your comment; we agree with the design of our study and we cannot fully establish a correlation between protection and total vaccine response.
We have synthesized the aim of our study: Given the need to implement vaccine strategies in these patients, our objective was to describe the serological and T-cell responses after the third dose of the vaccine in a cohort of patients with IMRDs (rheumatoid arthritis and spondyloarthropathies) treated with immunosuppressive therapy (csDMARDs, bDMARDs and tsDMARDs). We aimed to identify the impact of these treatments on vaccine response and determine which patients would benefit from various vaccination strategies. We have made the correspondent changes in the introduction section.
3. In Table 1, additional treatments are listed, which are not represented in the figures (e.g. Mycophenolate). (How) were these treatments included in the analyses?
Reviewer: Only partially accepted. Give this information in the manuscript.
Thank you for your appreciation. We have added to the statistical analysis in the materials and methods section.

Reviewer 4 Report
Thank you for submitting the revised manuscript entitled " Specific cellular and humoral response after the third dose of anti-SARS-CoV-2 RNA vaccine in patients with IMRDs on biologic therapy" I appreciate your effort in addressing the concerns and questions raised in my initial review. Your detailed explanations and clarifications have significantly improved the manuscript. I look forward to seeing it published.
Author Response
Thank you so much for this comment. We appreciate your careful reading and your insightful interpretation of our results and suggestions.
Round 3
Reviewer 3 Report
The reviewer's comments/points of criticism of the second round of review have been answered in an acceptable manner.
English language fine
Author Response

(The authors gave the same response as above.)
